# Exploring the similarities and differences of variables collected by burn registers globally: protocol for a data dictionary review study

Emily Bebbington [1,2] Joanna Miles,[3] Michael Peck,[4,5] Yvonne Singer,[6] Ken Dunn,[7] Amber Young [8,9]

This protocol was presented at the 21st Congress of the International Society for Burn Injuries, 28th August – 1st September 2022, Guadalajara, Mexico.

For numbered affiliations see end of article.

**Correspondence to**
Dr Emily Bebbington;
e.bebbington@bangor.ac.uk

## ABSTRACT

**Introduction** Burn registers can provide high-quality clinical data that can be used for surveillance, research, planning service provision and clinical quality assessment. Many countrywide and intercountry burn registers now exist. The variables collected by burn registers are not standardised internationally. Few international burn register data comparisons are completed beyond basic morbidity and mortality statistics. Data comparisons across registers require analysis of homogenous variables. Little work has been done to understand whether burn registers have sufficiently similar variables to enable useful comparisons. The aim of this project is to compare the variables collected in countrywide and intercountry burn registers internationally to understand their similarities and differences.

**Methods and analysis** Burn register custodians will be invited to participate in the study and to share their register data dictionaries. Study objectives are to compare patient inclusion and exclusion criteria of each participating burn register; determine which variables are collected by each register, and if variables are required or optional, identify common variable themes; and compare a sample of variables to understand how they are defined and measured. All variable names will be extracted from each register and common themes will be identified. Detailed information will be extracted for a sample of variables to give a deeper insight into similarities and differences between registers.

**Ethics and dissemination** No patient data will be used in this project. Permission to use each register's data dictionary will be sought from respective register custodians. Results will be presented at international meetings and published in open access journals. These results will be of interest to register custodians and researchers wishing to explore international data comparisons, and countries wishing to establish their own burn register.

## INTRODUCTION

Approximately 9 million people globally sustain burn injuries each year requiring medical treatment, of whom 120 000 die.[1] Over 80% of these injuries and deaths occur in low- and middle-income countries.[1] Lower

## STRENGTHS AND LIMITATIONS OF THIS STUDY

⇒ Training and pilot exercises will be undertaken to ensure that register design data extraction is completed to a high standard.
⇒ Extracted information will be verified by a second researcher.
⇒ A custodian from each register will be invited to be part of the study team to ensure accurate interpretation of the data dictionaries, subsequent analyses and write up.
⇒ Registers will be identified from the peer-reviewed literature, which may miss those that have not published their findings.
⇒ The majority of burn registers exist in high-income countries, so this study may underrepresent variables important for burn prevention and care in lower resourced environments.

income countries often have poor coverage of surveillance data, meaning that the true burden of disease in these countries is not fully known.[1 2] Where data are collected, it frequently does not include information required to inform prevention and intervention strategies, such as disaggregation of data by injury intent in regions where there are high rates of deliberate burns.[1 3–6] Burn mortality surveillance statistics are compiled from civil registration and vital statistic data, whereas burn morbidity statistics are calculated from hospital-based data.[2]

Burn registers provide clinical data that can be used for international morbidity surveillance. However, few international burn data comparisons are completed beyond basic morbidity and mortality statistics. Registers can be used for outcome assessment, research, planning service provision, clinical governance, quality improvement, service accreditation or, as clinical quality registers, to identify variation in practice.[7] The utility of burn registers is such that there are now

numerous countrywide registers (eg, Dutch Burn Repository) as well as intercountry registers (eg, the Burn registry of Australia and New Zealand collects data from Australia and New Zealand, the International Burn Injury Database collects data from England and Wales, the German Burn Registry collects data from German speaking countries and the American Burn Association's Burn Care Quality Platform Registry collects data from US centres and some international burn centres).[8–12] Established burn registers are strongly concentrated in high-income countries, most likely due to ethical, regulatory, technological and economic issues.[13] A notable exception is the WHO Global Burn Registry (WHO GBR).[14] This register allows any healthcare facility globally to submit and analyse their data for free. Twenty countries submit data to the WHO GBR, the majority of which are middle-income countries.[15] Most countries that submit data do not have a countrywide burn register. The success of the WHO GBR in countries without an active burn register likely reflects the enthusiasm of the international burn community for rigorously collected and collated burn injury data.

The variables collected by burn registers are not standardised internationally, thereby limiting international data comparisons. The development of a set of variables that are collected across all registers in a standardised way (an international minimum data set) would allow the comparison of data on issues of international significance. Pooling data from registers effectively achieves a larger sample size allowing investigation of rarer exposures and outcomes, tracking of emerging trends, investigation of how disease processes are affected by sociocultural factors and embedding trials.[16–21] Custodians of a countrywide or intercountry register might choose to incorporate an international minimum data set into existing data collection processes to help facilitate international data comparisons. but are likely to continue to collect country-specific variables required to tailor prevention strategies, service provision and quality improvement to the long-term needs of their population.

It is not known whether burn registers already collect any variables in a way that would allow international data comparisons. If data are not comparable, an international minimum data set would need to be developed as differences between registers may represent sources of bias during analyses. To achieve this will require internationally agreed variable definitions and methods of measurement. It is necessary to understand which variables are commonly collected across all registers prior to the development of an international minimum data set as it is likely that important common themes at present may not be collected in comparable ways. Agreed definitions would also be helpful to countries wishing to establish their own burn registers. Little work has been done to understand the similarities and differences across burn registers internationally. The aim of this project is to compare the variables collected in countrywide and intercountry burn registers internationally to understand their similarities and differences.

## METHODS AND ANALYSIS
The study objectives are to:
1. Compare patient inclusion and exclusion criteria of each participating burn register.
2. Determine what variables are collected by each register and if variables are required or optional.
3. Identify whether any variables are collected by all registers and identify common variable themes.
4. Undertake a detailed comparison of a sample of variables to understand differences in definitions and measurement methods.

The steps of the study are shown in figure 1. No reporting guidelines for protocols or studies of this nature were found on the Equator Network website (www.equator-network.org). Any deviations from the study protocol will be reported in the results manuscript.

### Terminology
Common terms applicable to the study have been defined to ensure uniformity of understanding across international collaborators (table 1). Standard definitions have been used where possible. These definitions will be used in all research materials and manuscripts.

### Eligibility criteria for burn register participation in the study
Countrywide and intercountry burn registers will be invited to participate in the study. Registers will be identified from a scoping review of active burn registers.[22 23] A register will be classified as countrywide or intercountry if there is the potential for healthcare facilities across a single country or multiple countries to submit data. The WHO GBR meets these inclusion criteria despite data submission being locally, rather than nationally, coordinated. The variables included in the WHO GBR are of particular importance because of the wide uptake of the WHO GBR in low- and middle-income countries, which are under-represented in burn register studies. Registers that are restricted to a single state or region of a country will be excluded in countries with an active countrywide or intercountry register. Burn register pilot studies will be included for countries that do not have an active burn register to attempt to further increase representation of low- and middle-income countries. Contact information from the register website will be used to invite register custodians to participate in the study. In cases where there is no register website, corresponding authors of recent register publications will be contacted to provide up to date information about the register custodian. Each custodian will be asked to provide a copy of their most recent data dictionary or equivalent document that explains which variables are collected by the burn register. Registers that have freely available data dictionaries will be automatically included in the study.

### Data handling and storage
No patient data will be collected. Data dictionaries and project documents will be held on an encrypted cloud storage system to allow international collaboration across

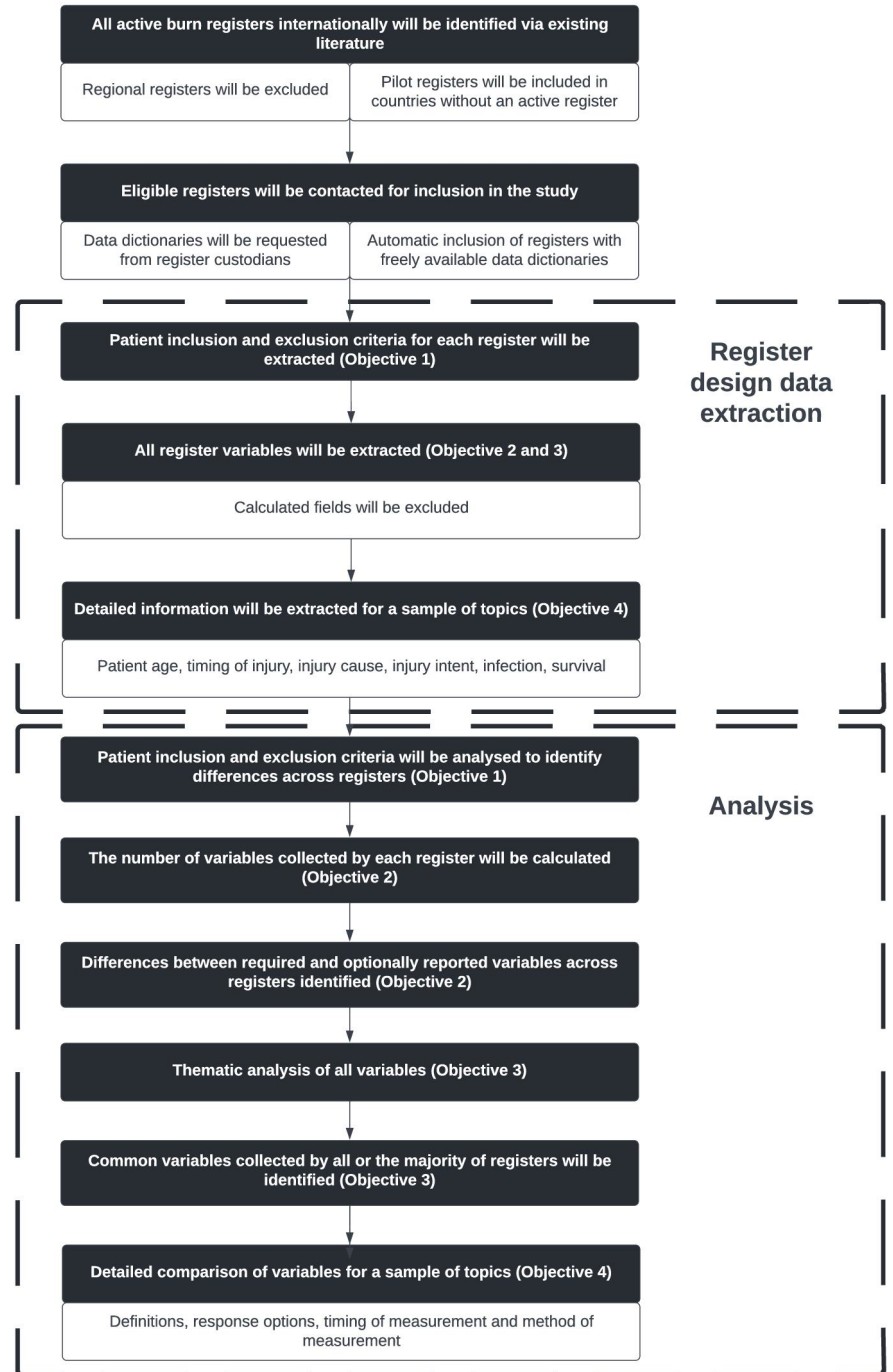

**Figure 1** The process of register recruitment, data extraction and data analyses that the study will follow.

institutional boundaries. Access to the cloud storage system will be agreed by the authors and permissions set accordingly.

### Register design data extraction rationale and process
#### Objective 1: Compare patient inclusion and exclusion criteria of each participating burn register
Each register uses a set of inclusion and exclusion criteria to determine which patients will have their information recorded in the burn register. For example, a register may include only patients with a burn injury requiring admission for more than 24 hours and exclude patients receiving care on an outpatient basis. Inclusion and exclusion criteria will be compared between registers to understand the patient populations under study. Inclusion criteria heterogeneity could represent a source of selection bias if data were compared without allowing for this. One author (EB) will extract patient inclusion and exclusion criteria from each register's data dictionary into a spreadsheet file. Where this information is not included in the data dictionary, it will be sought from the register website or custodian (order of preference). All of the data will be verified by a second author (JM).

**Table 1** Definitions of key terms used in this study

| Term | Definition | Example |
|---|---|---|
| Registry | An organisation and associated systems that support the upkeep of a register.[7] | WHO Global Burn Registry |
| Register custodian | Organisation responsible for maintaining reliability and security of a register's data.[26] | WHO |
| Register | A physical or electronic collection of pre-specified and systematically recorded details.[7] | Details about burn presentations to a hospital |
| Data dictionary | Document that defines each variable in a register, their limits, and validation parameters.[27–29] It does not include any patient data. | Specific to each register. May use standard definitions for a variable such as International Classification of Diseases 11th Revision. |
| Variable | One feature of interest in a register. | 'Date of birth', 'total body surface area of burn' (TBSA), 'discharge disposition'. |
| Variable response options | Potential choices to answer a variable. | Response options for 'discharge disposition' may include 'discharged home', 'transferred', 'discharged against medical advice', 'died'. |
| Required variable | A variable that must be inputted by the person completing data entry. | These are specific to each register and are dependent on the analyses that are deemed essential. Essential variables might include 'date of birth', 'TBSA'. |
| Optional variable | A variable that is not required to be collected about every patient. | These are specific to each register and are dependent on the analyses that are deemed important but not essential. For example 'income', 'occupation'. |
| Minimum data set | The list of required variables collected by the register | Specific to each register. |
| Electronic database | Collection of data organised for rapid search and retrieval by a computer.[30] | Structured Query Language database. |
| Calculated field | A piece of information that is computed using variable data. | Age at the time of the injury may be calculated using the variable 'date of birth' and 'date of injury'. |
| Outcome | Variable measured at a specific time point to assess the efficacy or harm of an intervention.[31] | Quality of life |
| Outcome measure | Method of quantifying an outcome of interest. | Quality of life can be measured using the EuroQol-5 Dimensions instrument. |

**Objectives 2 and 3: Determine what variables are collected by each register and if variables are required or optional. Identify whether any variables are collected by all registers and identify common variable themes.**

A diverse range of required and optional information is collected about patients in burn registers. We will collate and compare required and optionally collected variables noting differences between them. Calculated fields will not be included because they are a form of analysis reflecting the expertise of the data analyst rather than the raw data collected by the register. All variable names from the data dictionaries will be extracted by a researcher (JM) and 100% verified by a second researcher (EB). Data will be extracted into a spreadsheet file. The same extracted data will be used for objectives 2 and 3.

**Objective 4: Undertake a detailed comparison of a sample of variables to understand differences in definitions and measurement methods.**

Data about the same topic may be collected by each register in different ways. For example, information on the intent of the burn injury may include multiple variables such as patient-reported injury intent, physician suspicion of injury intent and International Classification of Diseases external causes of morbidity codes. Each register may use different definitions for the variable and include different response options. Comparison of data that has been collected using different definitions, methods or timing of measurement would represent a potential source of misclassification bias.

Detailed information will be extracted from the data dictionaries for a sample of topics. These will include 'patient age', 'timing of injury', 'injury cause', 'injury intent', 'infection' and 'survival'. The topics are chosen because they are likely to be collected by all registers. All variables related to each topic will be extracted. This will allow more comprehensive comparisons of variables across registers.

A pilot exercise will be completed in two phases to ensure that the detailed variable information is extracted accurately. First, two researchers (EB/JM) will extract detailed information on a sample of 20 variables from two freely available data dictionaries. This will allow the development and refinement of the data dictionary extraction form—a spreadsheet file with predefined column headings (eg, variable name, variable definition, method of measurement). Codes will be developed to ensure that a reason is assigned for missing fields in the extraction form. Second, detailed variable extraction for the topics 'patient age' and 'timing of injury' will be completed independently by two researchers (EB/JM) using the

extraction form. Inter-rater reliability will be calculated. Providing a good level of agreement is reached (Kappa statistic >0.60), data dictionary extraction will then be split equally between the two researchers (EB—injury cause and injury intent, JM—infection and survival). Regular discussion will be held to ensure any method developments are documented and applied universally. These will be reported in the final paper.

## Analysis, synthesis, and presentation
A custodian from each register will be invited to be part of the study team to ensure accurate interpretation of the data dictionaries, analyses and write up.

### Objective 1: Compare patient inclusion and exclusion criteria of each participating burn register
A table will be presented in the main manuscript that includes the register name, countries contributing to the register and patient inclusion and exclusion criteria.

### Objective 2: Determine what variables are collected by each register and if variables are required or optional
A full list of variables will be presented for each register with required and optional variables reported differently. Summary data on the number of variables (required and optional) collected by each register will be presented in tabular format in the main manuscript to understand the differences in volume of data collection occurring internationally.

### Objective 3: Identify whether any variables are collected by all registers and identify common variable themes
Variables will be divided into iteratively developed clinically meaningful thematic groups (eg, injury causation and severity) and subgroups (eg, injury intent, size of burn) by two researchers (EB/JM) and checked by a third (YS). Variables in each group and subgroup will be compared to identify the most commonly collected variable themes across registers and if any variables are collected by all registers in the same way. These will be listed in the manuscript. Common themes across registers will be presented as a figure.

### Objective 4: Undertake a detailed comparison of a sample of variables to understand differences in definitions and measurement methods
Detailed information collected by each register for variables relating to the topics 'patient age', 'timing of injury', 'injury cause', 'injury intent', 'infection' and 'survival' will be compared across registers and presented in tables. Comparisons will be made about the variable definitions, response options, timing of measurement and method of measurement.

## Patient and public involvement
A member of the public with experience of burns service planning and commissioning has reviewed this manuscript for readability and to ensure the needs of the service user are represented.

## ETHICS AND DISSEMINATION
This project involves the analysis of burn register data dictionaries and freely available burn register-related documents (eg, website, research papers) only. No human participant data will be used in this project. Therefore, ethical approval is not required in accordance with UK Research and Innovation and the Declaration of Helsinki.[24 25] Permissions will be requested from each register custodian to analyse data dictionaries where these dictionaries are not freely available. Only data dictionaries that we have express permission to use will be included in the project. Data will be anonymised and aggregated as required. The identity of the register will only be given with permission of the custodian. Prospective registration has not been completed as there is no appropriate register for this study type.

Results will be presented at international academic meetings to reach interested stakeholder groups (eg, International Society for Burn Injuries World Congress). Peer-reviewed publications of results will be published open access where possible to ensure accessibility. Custodians will be encouraged to disseminate the results in their country or territory. The collaboration formed for this study will be the basis for working together to address any recommendations for future work from the results manuscript.

**Author affiliations**
[1]Centre for Mental Health and Society, Bangor University, Bangor, UK
[2]Emergency Department, Ysbyty Gwynedd, Bangor, UK
[3]Plastic and Reconstructive Surgery Department, Norfolk and Norwich University Hospitals NHS Foundation Trust, Norwich, UK
[4]Arizona Burn Center, Valleywise Health Medical Center, Phoenix, Arizona, USA
[5]Department of Surgery, Creighton University Health Sciences Campus, Phoenix, Arizona, USA
[6]Victoria Adult Burn Service, The Alfred Hospital, Melbourne, Victoria, Australia
[7]Burn Care Informatics Group, NHS England, Manchester, UK
[8]Children's Burn Research Centre, University Hospitals Bristol and Weston NHS Foundation Trust, Bristol, UK
[9]Bristol Centre for Surgical Research, Population Health Sciences, Bristol Medical School, University of Bristol, Bristol, UK

**Acknowledgements** Sadly, Professor Amber Young passed away in the time between submission and publication of this paper. The co-authors would like to acknowledge her tremendous contribution to this project and the burn community in general. The authors also thank Mr Roy Dudley-Southern MBE for his time reviewing the manuscript and his useful comments.

**Contributors** EB, JM, MP, YS, KD and AY conceived the study and refined the study methods that are presented in this protocol manuscript. EB wrote the first draft of the manuscript. All authors reviewed the manuscript and agreed on the final version. EB and KD act as guarantors of the work.

**Funding** This article presents independent research funded by the UK National Institute for Health Research (NIHR) Advanced Research Fellowship NIHR 301362. The views expressed are those of the authors and not necessarily those of the United Kingdom National Health Service, the NIHR or the United Kingdom Department of Health. The study was also supported by the NIHR Biomedical Research Centre at the University Hospitals Bristol and Weston NHS Foundation Trust and the University of Bristol.

**Competing interests** EB, JM, MP and YS do not declare any conflicts of interest. KD is the medical director of the International Burn Injury Database (iBID). As part of this role he is the co-chair of the Burn Care Informatics Group. This is a part time role funded by NHS England. AY is funded to undertake a priority setting exercise

and a registry-based trial with the iBID registry. This work is part of work funded by the National Institute of Health Research Advanced Fellowship awarded to Dr Amber Young (Ref NIHR301362). The views expressed are those of the author(s) and not necessarily those of the NIHR or the Department of Health and Social Care. AY has received funding for a cohort study on the impact of genetic make-up on scarring after small area scalds from the Scar Free Foundation. AY is a minority stakeholder in a small-medium sized enterprise on burn wound infection. She has not received any money or expenses from this.

**Patient and public involvement** Patients and/or the public were involved in the design, or conduct, or reporting, or dissemination plans of this research. Refer to the Methods section for further details.

**Patient consent for publication** Not applicable.

**Provenance and peer review** Not commissioned; externally peer reviewed.

**Data availability statement** Not applicable.

**ORCID iDs**
Emily Bebbington http://orcid.org/0000-0003-1332-7558
Amber Young http://orcid.org/0000-0001-7205-492X

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
