## [Reviewer comments · BMJ Open]

ARTICLE DETAILS

TITLE (PROVISIONAL)	EXPLORING THE SIMILARITIES AND DIFFERENCES OF VARIABLES COLLECTED BY BURN REGISTERS GLOBALLY: PROTOCOL FOR A DATA DICTIONARY REVIEW STUDY
AUTHORS	Bebbington, Emily; Miles, Joanna; Peck, Michael; Singer, Yvonne; Dunn, Ken; Young, Amber

VERSION 1 – REVIEW

REVIEWER	Nikkia Allorto University of KwaZulu-Natal, Surgery
REVIEW RETURNED	06-Oct-2022

GENERAL COMMENTS	This is a well defined and presented protocol for a study of value to the burn community
--

REVIEWER	Vikash Ranjan Keshri The George Institute for Global Health, Injury Division
REVIEW RETURNED	17-Nov-2022

GENERAL COMMENTS	I must commend the authors for this important work. The epidemiology of burns, especially in low and middle-income countries (LMICs), is not understood and the registry has the potential to fill some of these gaps. My concerns here are about the limited scope of this study. The study objectives are limited to just understanding the variables from the data dictionary, which may well highlight the differences, but this must be seen in the light of overall objectives and uses of any burn registry systems. As most of the burn registries are not representative, the regional or country comparison based on a facility-based burns registry will probably not be appropriate. Burn registries in different settings and led by different entities have different objectives. It would be more appropriate for the resource involved in the study to try to document the technical and operational challenges of the such registry and their current use and utility. I think supplementing the qualitative inquiry with leaders/custodians. As rightly identified by the author's team, the current use of data compilation for burns from LMIC will not be captured, so the global comparison. WHO GBR is not country but facility specific? Even the WHO global burn registry is voluntary, and not all burn centre in each of the participating country is participating. Thus they cannot be termed as country burn registry. The author team is all from one HIC country; such a global research initiative would benefit from the diversity of authors from different
--

	regions and income groups who may provide contextual understanding. Deciding to include a variable in the burn registry is also context specific In the definition of the 'Register' – both physical and electronic – how to find about paper/register-based registry. Further, 'Registry' is defined as 'an organisation which supports the systems of delivery – must include or else a hospital maintaining registry would be registered'. By this definition, even a single burn unit maintaining data in paper-based registry systems qualifies to be a burn registry. Will they be included in the exercise, and if yes, what efforts would be made to search all such registry? The inclusion criteria talk about country-wide and inter-county registries but burns registries are often not country wise, with many burn centres not participating. For example, in WHO GBR, only a few centres from many countries participate. At the same time, inclusion criteria mention 'pilot registry will be included' while a single state or region will be excluded. I believe most 'pilot burn registries' will be either in a single centre of the region or only a few centres. Not sure if the data dictionary will have information on inclusion and exclusion criteria. Overall, I suggest thinking about what best can be done with the resources and time involved in this study. Thank you
--	--

REVIEWER	Doha Obed Hannover Medical School
REVIEW RETURNED	18-Dec-2022

GENERAL COMMENTS	The authors propose a well composed and comprehensive protocol, which seeks to address an important clinical issue. Studies like these, which evaluate large-scale burn data, should be encouraged. Most mentioned aspects are well-considered and well thought out. I have two points, which I hope the authors will be able to address:  1. Some burn registers provide comprehensive national data, whereas other registers only display select data from few participating burn centers. Will the authors in any form account for this contrast in the representation of the burn populations in the included registers? 2. Country-wide and inter-country burn registers will be identified from a scoping review of active burn registers. As this review was published in June 2022, it would not be considered outdated. However, the authors may miss burn registers with an inception after the review's publication. Will they consider further sources for the identification of newer registers?
--

VERSION 1 – AUTHOR RESPONSE

Reviewer: 1 Dr. Nikkia Allorto, University of KwaZulu-Natal, Grey's Hospital

Reviewer comment: This is a well defined and presented protocol for a study of value to the burn community

Author response: We thank Dr Allorto for her positive review.

Reviewer: 2 Dr. Vikash Ranjan Keshri, The George Institute for Global Health, University of New South Wales Faculty of Medicine

Reviewer comment: I must commend the authors for this important work. The epidemiology of burns, especially in low and middle-income countries (LMICs), is not understood and the registry has the potential to fill some of these gaps.

Author response: We thank Dr Keshri for his comments. Based on the last part of this comment, we are unsure if Dr Keshri interprets that in this paper we are proposing to create a new registry. We wish to clarify this is not the aim of this study. The aim of this project is to understand variables collected by existing burn registers from around the world. Particularly, to understand their similarities and differences such that we can anticipate challenges and identify limitations if data were compared in the future. We are not proposing that any patient data from the registers is compared in this project. We have added clarifying comments throughout the manuscript to make it clear the project will be comparing data dictionaries only – not patient data.

Reviewer comment: My concerns here are about the limited scope of this study. The study objectives are limited to just understanding the variables from the data dictionary, which may well highlight the differences, but this must be seen in the light of overall objectives and uses of any burn registry systems.

Author response: The wider and important issues of collection of burns data globally is out of scope for this specific project. We believe nonetheless, that understanding the variables currently collected by burn registers internationally will be of interest and value to the burn community. This is an essential first step in any future collaborations that aim to compare register data. This study will identify what current data in different registers can and cannot be compared. As registers undergo dataset revisions, this study will provide a roadmap so that the registers can align more closely in the future. We have kept the objectives of this project deliberately achievable such that they can be delivered in a timely manner. It is likely that once we have published the results of this study it will lead to further inter-registry collaborations.

Reviewer comment: As most of the burn registries are not representative, the regional or country comparison based on a facility-based burns registry will probably not be appropriate. Burn registries in different settings and led by different entities have different objectives. It would be more appropriate for the resource involved in the study to try to document the technical and operational challenges of the such registry and their current use and utility. I think supplementing the qualitative inquiry with leaders/custodians.

Author response: We agree that a study investigating the technical and operational challenges for burn registers globally is an important piece of work, and that a qualitative method would be appropriate to investigate this. However, the objective of this project is to investigate the similarities and differences of the variables collected by burn registers internationally in order to understand their potential for comparison in the future. Given the different objective of our study, we feel our proposed methods are appropriate.

Reviewer comment: As rightly identified by the author's team, the current use of data compilation for burns from LMIC will not be captured, so the global comparison. WHO GBR is not country but facility specific? Even the WHO global burn registry is voluntary, and not all burn centre in each of the participating country is participating. Thus they cannot be termed as country burn registry.

Author response: We thank Dr Keshri for drawing it to our attention that it may not be clear in the methods section that the WHO GBR would meet the inclusion criteria for the study, and the rationale for this. A register will be classed as country-wide or inter-country, and therefore invited to take part in the study, "if there is the potential for healthcare facilities across a single country or multiple countries to submit data". Any healthcare facility can submit data to the WHO GBR and therefore would meet our inclusion criteria. We recognise that these data are unlikely to be representative of an entire country, however given the wide uptake of the WHO GBR globally, particularly in LMICs which are underrepresented in burn register studies generally, we feel its inclusion is important as the variables

are in use on a large scale. We have added detail to the eligibility criteria section to make this clearer to the reader:

“Country-wide and inter-country burn registers will be invited to participate in the study. Registers will be identified from a scoping review of active burn registers.^{28,29} A register will be classed as country-wide or inter-country if there is the potential for healthcare facilities across a single country or multiple countries to submit data. The WHO GBR meets these inclusion criteria despite data submission being locally rather than nationally coordinated. The variables included in the WHO GBR are of particular importance because of the wide uptake of the WHO GBR in low- and middle-income countries, which are underrepresented in burn register studies.”

Reviewer comment: The author team is all from one HIC country; such a global research initiative would benefit from the diversity of authors from different regions and income groups who may provide contextual understanding. Deciding to include a variable in the burn registry is also context specific
Author response: The author team are from three HICs - the UK, Australia, and United States. We have invited register custodians from all participating registers to be authors in the results manuscript to try to maximise diversity and provide contextual understanding to the findings.

Reviewer comment: In the definition of the ‘Register’ – both physical and electronic – how to find about paper/register-based registry. Further, ‘Registry’ is defined as ‘an organisation which supports the systems of delivery – must include or else a hospital maintaining registry would be registered’. By this definition, even a single burn unit maintaining data in paper-based registry systems qualifies to be a burn registry. Will they be included in the exercise, and if yes, what efforts would be made to search all such registry?

Author response: We agree that there are likely to be burn centres that maintain paper-based records used for register purposes. Unfortunately, it is not feasible to identify and invite all single centre registers to be part of this project. We chose to focus on country and inter-country registers to maximise our understanding of the types of variables being collected in burn registers internationally.

Reviewer comment: The inclusion criteria talk about country-wide and inter-county registries but burns registries are often not country wise, with many burn centres not participating. For example, in WHO GBR, only a few centres from many countries participate.

Author response: Please see point above about the rationale for including the WHO GBR in this project. Please note in the inclusion criteria we define a country-wide or inter-country register as one in which “there is the potential for healthcare facilities across a single country or multiple countries to submit data”. We are not suggesting that the data collected by such registers is representative of the entire population of that country.

Reviewer comment: At the same time, inclusion criteria mention ‘pilot registry will be included’ while a single state or region will be excluded. I believe most ‘pilot burn registries’ will be either in a single centre of the region or only a few centres.

Author response: We highlight in the protocol that pilot registers identified in the scoping review of burn registers would be invited if they were from a country that does not currently have an active burn register. The rationale for this was to try to increase representation of countries without a national register. We agree that pilot registers are likely to only operate in a few centres, however there is still valuable information to be learnt about which variables can be collected and are felt to be important in that setting, and therefore add greater understanding to this project. The relevant section of the manuscript is as follows:

“Registers that are restricted to a single state or region of a country will be excluded in countries with an active country-wide or inter-country register. Burn register pilot studies will be included for

countries that do not have an active burn register to attempt to further increase representation of low and middle-income countries.”

Reviewer comment: Not sure if the data dictionary will have information on inclusion and exclusion criteria.

Author response: We highlight in the protocol that if this information is not in the data dictionary it will be sought from the register custodian, website, or publications. The relevant section of the manuscript is as follows:

“One author (EB) will extract patient inclusion and exclusion criteria from each register’s data dictionary into a spreadsheet file. Where this information is not included in the data dictionary, it will be sought from the register website or custodian (order of preference).”

Overall, I suggest thinking about what best can be done with the resources and time involved in this study.

Thank you

Reviewer: 3 Dr. Doha Obed, Hannover Medical School

Reviewer comment: The authors propose a well composed and comprehensive protocol, which seeks to address an important clinical issue. Studies like these, which evaluate large-scale burn data, should be encouraged. Most mentioned aspects are well-considered and well thought out. I have two points, which I hope the authors will be able to address: Some burn registers provide comprehensive national data, whereas other registers only display select data from few participating burn centers. Will the authors in any form account for this contrast in the representation of the burn populations in the included registers?

Author response: We thank Dr Obed for their comments. We agree that the scope of the population captured by each register may differ and could create bias should these populations be compared in the future. However, in this study we are only comparing the characteristics of the registry data items and will not be analysing patient data per se. Nonetheless, we will capture some of this information in the inclusion and exclusion criteria for each register, and we will also present the countries and centres covered by each register. Further exploration of differences in data coverage would likely require qualitative enquiry with register custodians and analysis of register patient data, thus are beyond the scope of this current project. We hope to explore these points in future work.

Reviewer comment: Country-wide and inter-country burn registers will be identified from a scoping review of active burn registers. As this review was published in June 2022, it would not be considered outdated. However, the authors may miss burn registers with an inception after the review’s publication. Will they consider further sources for the identification of newer registers?

Author response: We agree that newly established burn registers, particularly those that are yet to publish their findings, are likely to be missing from the scoping review and thus not included in this project. We have highlighted this as a limitation at the start of the manuscript: “Registers will be identified from the peer reviewed literature, which may miss those that have not published their findings.” We hope, however, that the publication of this project protocol and the subsequent results paper will raise awareness about this international registry work and encourage other registries to become involved. Presentation of this protocol at the International Society for Burns Injuries Congress led to two custodians of burn registers that are being established (Latin American and Belgian) approaching the presenter. The project is likely to lead to further burn registry comparison work (e.g. development of a minimum data set), and participation of all register custodians will be welcomed.

VERSION 2 – REVIEW

REVIEWER	Vikash Ranjan Keshri The George Institute for Global Health, Injury Division
REVIEW RETURNED	31-Jan-2023
GENERAL COMMENTS	Thank you for clarifying on the issues raised in earlier version.